# Fungicidal Activity of Recombinant Javanicin against *Cryptococcus neoformans* Is Associated with Intracellular Target(s) Involved in Carbohydrate and Energy Metabolic Processes

**DOI:** 10.3390/molecules26227011

**Published:** 2021-11-20

**Authors:** Santhasiri Orrapin, Sittiruk Roytrakul, Narumon Phaonakrop, Siriwan Thaisakun, Khajornsak Tragoolpua, Amornrat Intorasoot, Suzanne McGill, Richard Burchmore, Sorasak Intorasoot

**Affiliations:** 1Division of Clinical Microbiology, Department of Medical Technology, Faculty of Associated Medical Sciences, Chiang Mai University, Chiang Mai 50200, Thailand; santhasiri.or@cmu.ac.th (S.O.); khajornsak.tr@cmu.ac.th (K.T.); 2Functional Proteomics Technology Laboratory, Functional Ingredients and Food Innovation Research Group, National Center for Genetic Engineering and Biotechnology (BIOTEC), Pathum Thani 12120, Thailand; sittiruk@biotec.or.th (S.R.); narumon.pha@biotec.or.th (N.P.); siriwan.tha@ncr.nstda.or.th (S.T.); 3Infectious Diseases Research Unit (IDRU), Faculty of Associated Medical Sciences, Chiang Mai University, Chiang Mai 50200, Thailand; 4Department of Microbiology, Faculty of Medicine, Chiang Mai University, Chiang Mai 50200, Thailand; amornrat.in@cmu.ac.th; 5Institute of Infection, Immunity & Inflammation and Glasgow Polyomics, College of Medical, Veterinary and Life Sciences, University of Glasgow, Glasgow G6 1BD, UK; suzanne.eadie@glasgow.ac.uk (S.M.); richard.burchmore@glasgow.ac.uk (R.B.)

**Keywords:** plant defensin, javanicin, *Cryptococcus neoformans*, antifungal action, proteomics

## Abstract

The occurrence of *Cryptococcus neoformans*, the human fungal pathogen that primarily infects immunocompromised individuals, has been progressing at an alarming rate. The increased incidence of infection of *C. neoformans* with antifungal drugs resistance has become a global concern. Potential antifungal agents with extremely low toxicity are urgently needed. Herein, the biological activities of recombinant javanicin (r-javanicin) against *C. neoformans* were evaluated. A time-killing assay was performed and both concentration- and time-dependent antifungal activity of r-javanicin were indicated. The inhibitory effect of the peptide was initially observed at 4 h post-treatment and ultimately eradicated within 36 to 48 h. Fungal outer surface alteration was characterized by the scanning electron microscope (SEM) whereas a negligible change with slight shrinkage of external morphology was observed in r-javanicin treated cells. Confocal laser scanning microscopic analysis implied that the target(s) of r-javanicin is conceivably resided in the cell thereby allowing the peptide to penetrate across the membrane and accumulate throughout the fungal body. Finally, cryptococcal cells coped with r-javanicin were preliminarily investigated using label-free mass spectrometry-based proteomics. Combined with microscopic and proteomics analysis, it was clearly elucidated the peptide localized in the intracellular compartment where carbohydrate metabolism and energy production associated with glycolysis pathway and mitochondrial respiration, respectively, were principally interfered. Overall, r-javanicin would be an alternative candidate for further development of antifungal agents.

## 1. Introduction

The number of immunocompromised individuals including patients with organ transplantations, AIDS, as well as people living with cancer and receiving chemotherapy is increasing worldwide [1]. These individuals are at risk of being infected with opportunistic fungal pathogens, resulting in the increase of morbidity and mortality rates [2]. The systemic mycoses linked with meningitis caused by *Cryptococcus neoformans* are being recognized as a major threat for immunocompromised patients, particularly those with HIV [3]. Treatment of cryptococcal meningitis relies on the use of three classes of anti-fungal drugs, amphotericin B, flucytosine, and fluconazole [4]. However, these agents are limited due to their adverse effects in which are sometimes lethal to humans [5,6]. In addition, the emergence of secondary resistance has become more difficult for treatment [7]. Hence, novel antifungal agents to control cryptococcosis are urgently needed.

Recently, antimicrobial peptides (AMPs) are considered as potential candidates for the treatment of infectious diseases [8]. Defensins, one of a diverse class of natural AMPs found in mammalian and plant cells, are small cysteine-rich peptides with multifaceted actions including host defense against microbial infection and immunomodulatory activity [9]. In humans, various defensins are identified whereas three β-defensins, termed β-defensin-1 (hBD-1), hBD-2, and hBD-3 exhibited fungicidal activity have been extensively elucidated and play a significant role in acute mucosal defense against *Candida albicans* [10]. 

In plants, defensins are one of several classes of antimicrobial peptides and most abundant in stomata and peripheral cells, where pathogens usually attack [11,12]. Plant defensins display remarkable broad-spectrum antimicrobial activity against bacteria, fungi, and viruses, albeit antifungal activity is commonly reported [11,13]. The common features of plant defensins including amphipathicity and cationicity enable them to permeabilize through biological membranes and pore-forming eventually caused of microbial cell death. Apart from direct action on the membrane, some plant defensins act as cell-penetrating peptides that exert their antimicrobial activity targeting specific intracellular compartments [14]. Javanicin is a small (approximately 6 kDa) defensin-type of antimicrobial peptide isolated from *Sesbania javanica* seeds. The peptide comprises of 47 amino acids with four disulfide bonds and is predicted to adopt a single alpha-helix and triple-stranded antiparallel beta-sheet structure. From our previous study, recombinant javanicin (r-javanicin) was produced using *E. coli* expression system [15]. The purified peptide was examined and exhibited potent antifungal activity against human fungal pathogens including *C. neoformans*. However, the mechanism of action of this anticryptococcal peptide is still unknown. Understanding the mode of action by which this molecule inhibits fungal growth is essential for the development of an antifungal therapeutic drug.

In this study, the antifungal effects of r-javanicin against *C. neoformans* were evaluated. The assessment of killing kinetics of the peptide was performed using a time-kill assay. The action of peptides on the fungal membrane was observed using a scanning electron microscope (SEM). The localization of r-javanicin was studied using both fluorescent and confocal laser scanning microscope (CLSM). Finally, cellular metabolic processes of *C. neoformans* after r-javanicin treatment were explored using proteomics analysis.

## 2. Results

### 2.1. Recombinant Avanicin Exhibited a Fungicidal Activity against C. neoformans

The antifungal properties of r-javanicin were examined using MIC assay. The results indicated that *C. neoformans* adjusted 10^6^ CFU/mL was completely killed at a peptide concentration of 25 µg/mL whereas it needed a two-fold increase (50 µg/mL) while 10^7^ CFU/mL yeast cells prepared for proteomic analysis, were examined. A time-kill of cryptococcal cells was conducted for 48 h and the substantial reductions of viable cells after treatment with various concentrations of peptide (25 (1 × MIC), 50 (2 × MIC) and 100 (4 × MIC) µg/mL) were observed (Figure 1). The growth inhibition of r-javanicin antimicrobial peptide against *C. neoformans* was initiated at 4 h post-treatment. Viable yeast cells gradually decreased after incubation with 1 × MIC concentration of peptide whereas *C. neoformans* were more rapidly killed when higher concentrations of peptide were examined. As a result, the killing activity of r-javanicin was mediated in a dose- and time-dependent manner (Figure 1a). At the MIC value, the growth inhibition of cryptococcal cells was reduced over 50% within 24 h and completely eradicated in 48 h (Figure 1b). Therefore, r-javanicin at 0.5 × MIC could not reduce cryptococcal cells and represented a log increase (2.75 ± 0.1656) comparable to the untreated group (2.71 ± 0.3100). The log reduction of cryptococcal cells after being treated with various concentrations of r-javanicin were shown in Table 1.

### 2.2. Javanicin Intracellularly Translocated into the Yeast Cell

Initially, SEM was employed for outer surface morphological observation of fungal cells after treatment with r-javanicin. *C. neoformans* incubated with a peptide suspension buffer was performed in parallel as a control group. The results indicated cells with a bright, spherical shape with a smooth surface and spiky protuberances in both control and r-javanicin treated samples. These results implied that the fungal membrane might not be a target of the peptide (Figure 2).

To investigate the localization of the peptide in cryptococcal cells, r-javanicin tagged with Flu-P1 was utilized for monitoring its subcellular localization by fluorescence microscope. Anti-GXM mAb and CFW specific for capsule and cell wall staining, respectively, were included in this study. The results showed that fungal capsule and cell wall could be located in both control and test groups whereas r-javanicin-Flu-P1 staining was only observed in fungal bodies of yeast cells treated with peptide (Figure 3). Additionally, CLSM was conducted and images defined the accumulation of the peptide in the cytoplasmic compartment (Figure 4). Overall, the data implied that r-javanicin is translocated across the membrane, targeting an intracellular compartment of *C. neoformans*.

### 2.3. Javanicin Interacts with Cellular Metabolic Pathways of C. neoformans

In this study, label-free quantitative shotgun proteomics analysis was employed for the determination of proteome of *C. neoformans* in response to r-javanicin. *C**ryptococcus neoformans* at inoculum of 10^7^ cells/mL were incubated with 50 µg/mL (MIC value) of r-javanicin for 0, 4, 8, 6 and 24 h. After analysis, a total of 2325 proteins were identified and a heat map of fungal proteins hits in various time points during peptide treatment compared to untreated control was picked off (Appendix A). Of these, 688 proteins were identified in the untreated control, 124 were individually identified in peptide treated group, and 1513 were expressed in overlap (Figure 5). One hundred and twenty-four proteins of *C. neoformans* in response to r-javanicin at various time points from 0–24 h were further analyzed and represented in the Venn diagram (Figure 6). It was found that 22 proteins were individually observed at the basal level (0-h incubation), therefore, these identified proteins were disappeared after peptide exposure. An upsurge of a total of 70 proteins consisting of 15, 22, 12, and 21 proteins were exclusively induced at 4, 8, 16, and 24 h posttreatment, respectively (Figure 6a). The proteins identified in each time point were further characterized using gene ontology analysis (Figure 6b, Appendix A). It was also observed that various fungal proteins expressions were typically induced at 4 h after peptide treatment in which a total of 15 proteins played role in 14 functions (Figure 6b) associated with DNA binding activity and catalytic activities that are responsible for cellular metabolic processes, including carbohydrate metabolism. Additionally, acetate kinase enzyme (A0A225XDF3) expression was commonly found in all conditions.

## 3. Discussion

New peptide discovery is increasing following the study of their diverse array of biological activities. Peptides with potent antimicrobial activity and limited toxicity to human cells enter into clinical research and development pipelines. Along the way, the study of the mechanism of actions of these candidate peptides has become one of the indispensable processes for further development for therapeutic potential. Javanicin, a class of defensin derived from the seed of *S. javanica*, was previously identified and heterologous expressed in *E. coli* [15]. The peptide exhibited potent anti-proliferative activity against breast cancer cells and antifungal activities in both human pathogenic yeast and mold, including fluconazole sensitive and fluconazole-resistant *Candida albicans*, *C. neoformans,* and *Trichophyton rubrum* [15]. However, the mode of action by which javanicin exerts its antifungal effect is still unknown. *C. neoformans*, a major causative agent of meningoencephalitis in patients with advanced AIDS was selected as a candidate for preliminary evaluating the mechanism of action of the antimicrobial peptide in this study. Regarding plant defensins, antifungal activity targeting the membrane surface followed by its permeabilization and/or intracellular biomolecules were previously described [16,17]. In our study, the time-kill of r-javanicin against *C. neoformans* was initially examined. Unlike the fungicidal activity of other defensins even completely killed yeast appears in minutes to a few hours [18,19], the activity of r-javanicin was dramatically low starting from 4 h to over 24 h. Based on the morphological change of cells, SEM is a primary tool for analyzing distinctive membrane morphology after peptide exposure [20,21]. SEM was conducted to study the morphology of the cryptococcal outer surface, therefore, negligible differences in outer surface with slight shrinkage was observed in r-javanicin treated cells. In fact, the internal cells alteration could not be seen when SEM was analyzed. The degree of cryptococcal cells damage after antimicrobial peptide exposure would be further investigated under transmission electron microscope (TEM). This technique has previously been applied for revealing more detail throughout the cells [22]. Whether r-javanicin renders the internal compartment(s) alteration of yeast cells would be evaluated.

Utilizing of fluorescent microscopy for the determination of the cellular localization of antimicrobial peptides has been effectively illustrated thus far [23]. Fluorescent labeled r-javanicin was prepared using the IPL system for peptide localization in *C. neoformans*. Anti-GXM-mAb and CFW specific for capsule and cell wall staining, respectively, were included in this experiment. The results indicated that the fluorescent intensity of antimicrobial peptides was detected intracellularly of yeast cells. The number of fungal cells incorporated with peptide was significantly increased when the incubation period was increased from 2 to 8 h (data not shown). The combination of SEM and CLSM analysis revealed that r-javanicin displayed fungicidal activity against *C. neoformans* through the intracellular target(s). Nevertheless, it is noted that the specific localization of peptides including cytoplasmic staining or colocalization study of existing cytosolic proteins in fungal cells would be further investigated to determine the exact location of peptides.

To better understand the intracellular interaction of peptides in yeast cells, time-course proteome analysis of *C. neoformans* in response to recombinant peptides was further undertaken. After incubation with its MIC (50 µg/mL) for 0, 4, 8, 16, and 24 h, the *C. neoformans* proteome was investigated by label-free based quantitative proteomics. Twenty-two proteins observed at basal level (0 h of incubation) were found and those were associated with catalytic activity, hydrolase activity, transporter activity, and transferase activity. Interestingly, a few of those were identified as virulence-associated proteins including ferric-chelate reductase (A0A225Y17) and antiphagocytic protein 1 (App1). Ferric-chelate reductase is an enzyme that catalyzes the reduction of bound ferric iron in iron chelators (siderophores). It is a key enzyme of the iron acquisition pathway, and thus critical for *C. neoformans* survival during iron-dependent growth [24]. Loss of this enzyme resulting in a defect of virulence factor production and the pathogenicity attenuation of *C. neoformans* has been mentioned [25]. App1 has been identified as a virulence factor by inhibiting the phagocytosis of host macrophages through complement receptors CR2 and CR3 [26]. However, these fungal proteins were disappeared when *C. neoformans* was incubated with r-javanicin for at least 4 h.

Treatment with r-javanicin for 4, 8, 16, and 24 h caused 70 unique proteins alteration in *C. neoformans*. They were responsible for 20, 16, 4, and 13 functions, respectively (Figure 6b). Those proteins were mainly associated with catalytic and nucleic acid binding activity (Appendix A). Noteworthy, 14 proteins induced by r-javanicin at 4 h were involved in 14 exclusive functions including nucleic acid binding (A0A225YTG2 and A0A225X7V2), transcriptional regulatory network (A0A225YAJ2), and the cellular response for surviving against r-javanicin [27,28]. Interestingly, it revealed that the majority of those upsurged proteins involved in catalytic activity were mostly responsible for variable metabolic processes. Many fungal intracellular proteins reflect the regulation of carbohydrate catabolism, such as glyceraldehyde-3-phosphate dehydrogenase (GPD) (A0A1Y0JXL3), pyruvate decarboxylase (PDC) (A0A225XTP6), 3-hydroxybutyryl-CoA dehydrogenase (HBD) (A0A225Y9Q8), and inositol oxygenase (IOX) (A0A225Y7C6). These enzymes are associated with either breaking down of some carbohydrates and energy production in the cell. The function of PDC has previously been described and plays a role in the fermentative process of yeast cells, especially in *Saccharomyces*, to produce ethanol [29].

Thus, concurrent induction of PDC indicated that carbohydrate flux may proceed through glycolysis into fermentation-like pyruvate. Likewise, IOX enzyme expression is required for inositol catabolism in response to low energy conditions of organisms [30]. In addition, a previous report indicated that inositol can be exploited as a carbon source in some fungal species, including *C. neoformans* [31]. Similarly, some catalytic enzymes are important for metabolic processes including nucleic acid binding (A0A225YK20 and A0A225YLC6), membrane transporter activity (A0A225YSN4), and transferase activity (A0A225YAB0), were identified at 8 h post-incubation. Although the activity of bis (5′-adenosyl)-triphosphatase (A0A225X9K2) in *C. neoformans* is still unknown, this enzyme-mediated apoptosis in human cancer cells has been reported [32]. Few potentially functional proteins were observed at the late stage (16 and 24 h post-incubation). Additionally, several expressed proteins with little known function were also observed at these time points. In this study, the presence of acetate kinase (AK) was found in each period of investigation time. AK enzyme also facilitates the generation of ATP from acetate through the central metabolic intermediate acetyl-CoA [33]. These pathways may include redundant proteins as a regulatory network to compensate themselves for existence during r-javanicin attack. The proteomic results of some fungal proteins alteration during r-javanicin activation were summarized in Figure 7.

*C. neoformans* is classified to be obligate aerobe [34]. The oxidative phosphorylation in mitochondria is the major source of ATP and essential for the survival of the fungal yeast cells [35]. Currently, it has been revealed that both AK and PDC are upregulated to convert pyruvate to acetaldehyde for acetate production under hypoxic condition [36,37]. The proteomics analysis of *C. neoformans* after javanicin exposure indicated the unique proteins expression for acetate production similar to that of oxygen-depleted. It is implied that javanicin might be involved in mitochondria’s function. Compared to the aforementioned peptides, few AMPs have been identified as a metabolic inhibitor thereby interfering with any enzymes related to cellular processes or effector molecules. Tu et al., (2011) reported that lactoferricin B inhibited bacterial growth affecting a number of metabolic pathways, and one action was highly related to pyruvate metabolism [38]. Whereas Bac-7 peptide caused a decrease in proteins involving in *E. coli* nucleic acid metabolism [39].

Overall, we hypothesized that r-javanicin affected a number of proteins involved in carbohydrate metabolism and impaired energy production. Interference of carbohydrate metabolism of *C. neoformans* would be related with polysaccharide-associated virulence factors alteration including fungal capsule [40]. So that the measurement of capsule thickness or determination of antifungal activity in capsular and acapsular strains in the presence of r-javanicin might be further performed for supporting the effect of peptide. In addition, our previous study revealed that r-javanicin induced the death of breast cancer cells (MCF-7) via apoptotic pathway [15]. Although it is still unclear about the correlation between the antifungal and anticancer activity of this novel peptide, it is possible that r-javanicin might directly target mitochondria where the induction of apoptosis is originated. loss of antibacterial activity of r-javanicin against both *E. coli* and *Staphylococcus aureus*, a candidate bacteria for gram-negative and gram-positive bacteria, respectively, was observed [15]. The bactericidal activity of the peptide can conceivably be abolished due to the lack of any intracellular target (mitochondria). Therefore, additional evidence is strongly needed to support this hypothesis. The final destination of r-javanicin following translocation across the biological membrane of eukaryotic cells needs further elucidation. The selective cellular compartment labeling is a powerful tool for determining cellular events such as mitochondrial staining. Together with the study on energy-relevant metabolic processes would be allocated for measuring the complexity of carbohydrate metabolism, including glycolysis, TCA cycle, as well as electron transport chain [41]. Ultimately, affinity proteomics based on the use of protein-specific detection has been currently introduced as a promising study of the endogenous complex. The particular strengths of this method including protein localization, functional characterization, and target identification are the impacts for the discovery of new therapeutic agents [42].

## 4. Materials and Methods

### 4.1. Microorganism

*Cryptococcus neoformans* H99 strain (serotype A) was kindly provided by Assoc. Prof. Dr. Pojana Sriburee, Department of Microbiology, Faculty of Medicine, Chiang Mai University, Chiang Mai, Thailand. The growth of pathogenic yeast was maintained on the Sabouraud dextrose agar (Thermo Fisher Scientific, Waltham, MA, USA), and incubated at 37 °C for 72 h. For the liquid culture, an isolated colony was cultivated in RPMI-1640 medium (Thermo Fisher Scientific, Waltham, MA, USA), incubated at 37 °C for 24 h with shaking. Yeast cells were harvested by centrifugation, washed twice with sterile phosphate-buffered saline; PBS (137 mM NaCl, 2.6 mM KCl, 10 mM NaH_2_PO_4_, 1.8 mM KH_2_PO_4_; pH 7.4), and the desired inoculum was quantified using the Neubauer chamber. In general, cells density at approximately 1 × 10^6^ cells/mL in RPMI-1640 medium supplemented with 2% glucose [43] were prepared in each experiment.

### 4.2. Recombinant Javanicin Production

In this study, r-javanicin was obtained according to the previous protocol [15]. Briefly, a mid-log phase culture of *E. coli* Origami 2 (DE3) strain carrying the pTXB1-javanicin plasmid was induced with 1 mM isopropyl β-D-1-thiogalactopyranoside (IPTG) and further incubated at 25 °C for 18 h with agitation. Bacterial cells were harvested by centrifugation at 4000× *g* at 4 °C for 10 min, washed thrice with PBS, and resuspended in B-PER^TM^ lysis reagent (Thermo Fisher Scientific, Waltham, MA, USA). To pellet insoluble material, the solution was centrifuged at 10,000× *g* for 10 min. The supernatant was collected and the recombinant fusion protein was further purified according to the manufacturer’s instruction (IMPACT™ kit; New England Bio Labs Inc., Ipswich, MA, USA). Cleavage of the fusion protein was induced by dithiothreitol (DTT) (Merck KGaA, Darmstadt, Germany). Finally, r-javanicin was dialyzed with stirred in PBS for 24 h, at 4 °C, and concentrated using NMWL 3 KDa ultracentrifugation (Amicon ultra 15 mL centrifugal filter; Merck KgaA, Darmstadt, Germany). The peptide concentration was measured, then checked on SDS-polyacrylamide gel electrophoresis and kept at −20 °C until determination.

### 4.3. Antifungal Activity of R-Javanicin against C. neoformans

Antifungal activity determination was performed according to the previous reports [44,45]. Two different inoculums of *C. neoformans* at 10^6^ and 10^7^ colony-forming units (CFU)/mL (the latter was provided for proteomic analysis) were prepared in a 96-well plate format. Fifty microliters of cell suspension in RPMI-1640 medium supplemented with 2% glucose was incubated with 50 μL of two-fold serially diluted peptide ranging from 0–200 μg/mL and further incubated at 37 °C for 48 h. The experiment was conducted in triplicate in three independent experiments, and the average values were determined.

### 4.4. Time-Kill Assay

The time-kill of r-javanicin against cryptococcal cells was evaluated according to the previously described protocol [44]. The total volume of 10 mL containing 5 mL of adjusted cell suspension and 5 mL of either peptide or PBS was set up. At each time point (0, 2, 4, 6, 8, 12, 16, 24, 36, 48 h), one hundred microliters of the reaction mixture were withdrawn and 10-fold serially diluted in PBS. Approximately 30 μL of diluted culture was spread on SDA plates and then were incubated at 37 °C for 48 h. The total number of yeast cells was counted, and the percentage of cell viability of *C. neoformans* versus time was plotted. The experiment was performed in triplicate independently three times, and the average values were determined.

### 4.5. Scanning Electron Microscopic Analysis

SEM was employed as previously described by Datta and colleagues [46]. The desired inoculum of yeast cells was incubated at 37 °C with the MIC value of r-javanicin at a final concentration of 25 μg/mL. Five hundred microliters of fungal suspension at 4 and 8 h post-treatment was aliquoted and then incubated with 2.5% glutaraldehyde at room temperature (RT) for 1 h. After fixation, the suspension was centrifuged at 5000× *g* for 5 min, the cell pellet was collected, washed, and resuspended in 20 μL phosphate buffer solution (0.0754 M Na_2_HPO_4_.7H_2_O, 0.0246 M NaH_2_PO_4_H_2_O; pH 7.4). Cells were placed on poly-L-lysine coated slides and air-dried for 1 h. Slides were gently washed with phosphate buffer solution and treated with 0.1 M of phosphate buffer containing 2% osmium for 1 h. After incubation, slides were washed once, air-dried for 30 min, and followed by the dehydration process using a series of ethanol. Untreated controls were assayed using the same protocol. Finally, the samples were gold-coated and observed under SEM (JEOL Ltd., Tokyo, Japan).

### 4.6. In Vitro Peptide Labeling with Flu-P1

To study the peptide localization in the yeast cell, r-javanicin was labeled with fluorescein dye. Flu-P1, a commercially available fluorescein synthetic peptide consisting of 7 amino acid residues (CDPEK(Fluorescein)DS), was kindly provided by Dr. Inca Ghosh, New England Biolabs, Inc. r-Javanicin was tagged with Flu-P1 fluorescent peptide through the intein-mediated protein ligation (IPL) system [47]. Following the manufacturer’s instructions, the purified recombinant peptide was diluted with 1×IPL reaction buffer to achieve the concentration of 1 mM. The peptide suspension was then mixed with 1 mM Flu-P1 at the ratio of 1:1, and further incubated at 23 °C for 4 hr. Fluorescent labeled r-javanicin was added into dialysis tubing with a 3.5 kDa molecular weight cut-off and dialyzed in PBS overnight at 4 °C. The labeled peptide was concentrated and kept at −20 °C until testing commenced.

### 4.7. Peptide Localization Using Fluorescence Microscopy

The reaction was modified from elsewhere [48]. An inoculum of *C. neoformans* was treated with r-javanicin-Flu-P1 at MIC value (25 µg/mL) and further incubation at 37 °C for 2 h and 8 h. The fungal suspension in each time point was collected and washed three times with PBS. After centrifugation at 5000× *g* for 5 min, yeast cells were then fixed with 4% paraformaldehyde. Fixed cells were blocked with PBS containing 2% human AB serum and further incubated on ice for 30 min. Monoclonal antibody (mAb) against glucoronoxylomannan (GXM) clone 18B7 (1:50) (kindly provided by Assoc. Prof. Dr. Sirida Youngchim, Department of Microbiology, Faculty of Medicine, Chiang Mai University, Thailand) was added and incubated on ice for 30 min. After washing, the mixture was further incubated with a 1:200 dilution of Alexa 568-conjugated goat anti-mouse IgG (Invitrogen, Carlsbad, CA, USA) followed by 20 µg/mL calcofluor white (CFW) (Merck KGaA, Darmstadt, Germany) for 30 min. Finally, the cell was pelleted and resuspended with anti-fade and observed under a fluorescent microscope and CLSM model LSM 980 (Carl Zeiss, Inc., Oberkochen, Germany). Control was done in parallel whereas r-javanicin-Flu-P1 was replaced by peptide suspension buffer.

### 4.8. Protein Preparation for Label Free Quantitative Proteomics

Protein preparation was conducted according to the previous report [49]. Briefly, adjusted *C. neoformans* cell suspension to 10^7^ CFU/mL was exposed to an equal volume of diluted peptide (a final concentration of peptide was 50 µg/mL). The mixture was incubated at 37 °C for 0, 4, 8, 16, and 24 h. Following incubation, the treated cells were withdrawn and collected by centrifugation at 5000× *g* for 5 min. The total cellular protein was extracted, quantified, solubilized with 0.5% SDS solution, and vortexed at RT for 1 h. The supernatant was collected by centrifugation at 10,000× *g* for 15 min. Two volumes of cold acetone was added and subsequently incubated at −20 °C overnight. The mixture was centrifuged, and the pellet was collected, air-dried, and then stored at −80 °C. After protein measurement, 5 µg of protein samples from each biological triplicate were subjected to an in-solution digestion process. Initially, protein pellets were completely dissolved in 10 mM ammonium bicarbonate (AMBIC). DTT solution (10 mM DTT in 10 mM AMBIC) was added into the mixture and further incubated at 60 °C for 1 h. The sample was alkylated using 15 mM iodoacetamide (IAA) in 10 mM AMBIC solution. The treated samples were gently mixed and further incubated in the dark at room temperature for 45 min. After removing the aIAA solution, protein samples were digested with sequencing grade trypsin (ratio 1:20) and incubated at 37 °C for overnight. Afterwards, digested peptides were dried using a speed vacuum concentrator, resuspended in 0.1% formic acid, and subjected for proteome analysis. The protein preparation was done in three independent experiments.

### 4.9. Liquid Chromatography-Tandem Mass Spectrometry (LC/MS) and Data Analysis

The processed samples were injected into a Hybrid Quadrupole Q-TOF Impact II LC-MS system (Bruker Daltonics Ltd.; Hamburg, Germany) coupled with a nanoLC system: UltiMate 3000 LC System (Thermo Fisher Scientific; Madison, WI, USA). Briefly, one microliter of digested proteins was enriched on a µ-Precolumn 300 µm i.d. × 5 mm C18 Pepmap 100, 5 µm, 100 A (Thermo Scientific, Nottingham, UK), separated on a 75 μm i.d. × 15 cm and packed with Acclaim PepMap RSLC, 2 μm C18, 100 Å, nanoViper (Thermo Scientific, UK). Chromatography was performed with solvent A (0.1% formic acid in water) and solvent B (0.1% formic acid in 80% acetonitrile). A gradient of 5–55% solvent B was used to elute peptides by a constant flow into a nanocolumn at flow rate of 300 nL/min. Electrospray ionization was carried out at 1.6 kV using the Captive Spray and nitrogen was used as a drying gas (flow rate about 50 L/h). Collision-induced dissociation (CID) product ion mass spectra were obtained using nitrogen gas for collision. Mass spectra (MS) and MS/MS spectra were obtained in positive-ion mode at 2 Hz over the range (*m*/*z*) 150–2200. The collision energy was adjusted to 10 eV as a function of the *m*/*z* value [50]. The LC-MS analysis of each sample was done in triplicate. For quantitative analysis, raw LC-MS data were analyzed by MaxQuant 1.6.6.0. with the Andromeda search engine to correlate MS/MS spectra to the Uniprot *C. neoformans* database [51]. The MaxQuant’s standard setting was as followed: maximum of two miss cleavages, a mass tolerance of 0.6 dalton for the main search, trypsin as a digesting enzyme, carbamidomethylation of cysteine as fixed modification, and the oxidation of methionine and acetylation of the protein N-terminus as variable modifications. Only peptides with a minimum of 7 amino acids, as well as at least one unique peptide, were required for protein identification. Only proteins with at least two peptides, and at least one unique peptide, were considered as being identified and used for further data analysis. Protein FDR was set at 1% and estimated by using the reversed search sequences. The maximal number of modifications per peptide was set to 5. Mass spectrometry proteomics data have been deposited to the ProteomeXchange Consortium via the jPOSTrepo [52] partner repository with the dataset identifier JPST001153 and PXD025806. 

## 5. Conclusions

In summary, the mechanism of action for r-javanicin against *C. neoformans* was preliminarily studied and likely implicated in intracellular target(s). The slow-killing kinetic represented by the peptide also strongly supported the role of the intracellular targeting mechanism and was involved in carbohydrate metabolism and energy production impairment. However, fundamental aspects of the specific pathways attacked by r-javanicin and the unique features on the cellular level remain a mystery. To what extent intracellular translocation of r-javanicin into cryptococcal cells to mediate fungal death for further development of fungal therapeutic drugs remains to be seen.

## Figures and Tables

**Figure 1 molecules-26-07011-f001:**
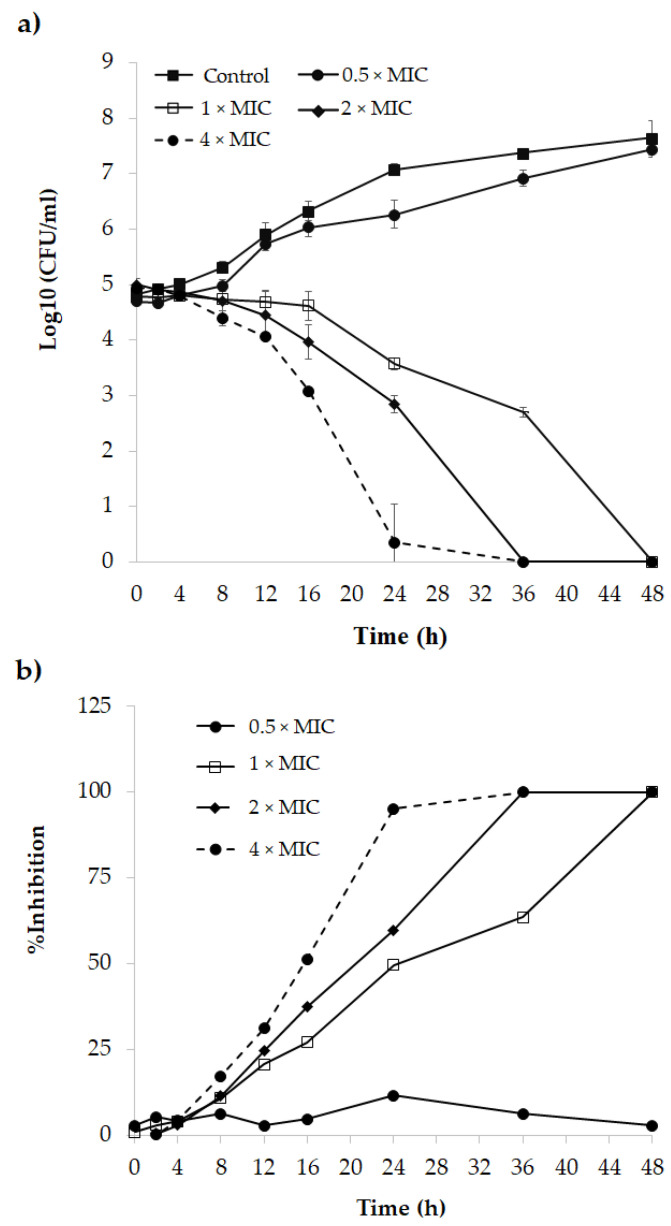
Time-killing profiles of r-javanicin against *C. neoformans*. (**a**) cell viability was determined by colony count and expressed in log 10 CFU/mL versus time. The error bars and standard deviation were represented in three independent experiments and performed in triplicate. (**b**) the percent inhibition calculated during fungal cells growth in the presence of various concentrations of r-javanicin compared with untreated cells was illustrated.

**Figure 2 molecules-26-07011-f002:**
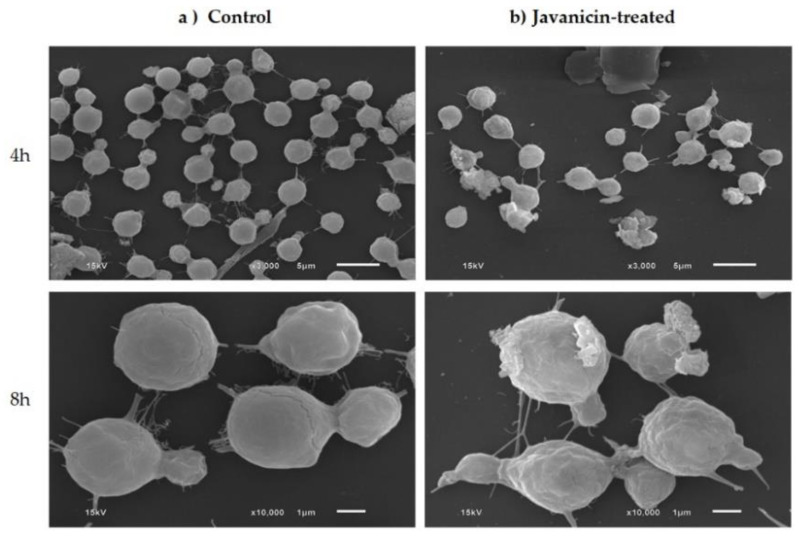
The illustration of SEM image segmentation of *C. neoformans* H99 strain in either untreated or r-javanicin treated. Panel (**a**) represented untreated cells control. Panel (**b**) indicated the yeast cells incubated with r-javanicin at its MIC value. Photographs were taken at 4 and 8 h after treatment. Bars at 4 and 8 h indicated 5 and 1 µm, respectively. There is negligible difference with slight shrinkage was observed in r-javanicin treated groups when compared to the untreated control.

**Figure 3 molecules-26-07011-f003:**
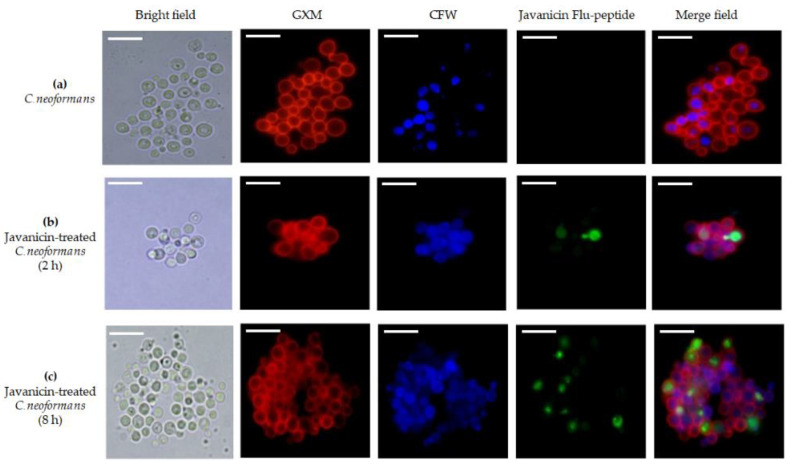
Fluorescence microscopic analysis of r-javanicin-Flu-P1 treated *C. neoformans*. Cryptococcal cells were incubated with peptide suspension buffer (untreated control) (**a**). *C. neoformans* was incubated with r-javanicin-Flu-P1 peptide at 2 (**b**) and 8 h (**c**), respectively. Photographs were taken at the same parameter settings. Red fluorescence represented the polysaccharide capsule staining with anti-GXM mAb followed by Alexa 568-conjugated goat anti-mouse IgG. Blue color showed the boundary of fungal cell wall stained by specific cellulose and chitin-binding dye, calcofluor white (CFW). Green fluorescence indicated the localization of antimicrobial peptide inside the yeast cells (scale bar = 10 µm).

**Figure 4 molecules-26-07011-f004:**
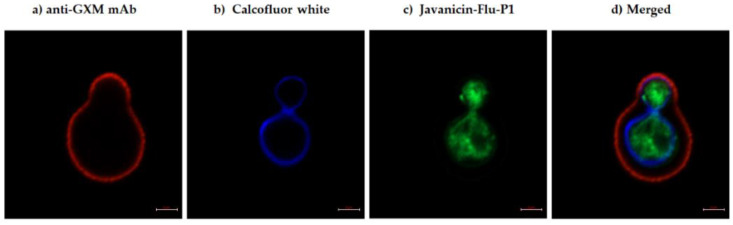
Fluorescent staining for r-javanicin localization in *C. neoformans* H99 using confocal laser scanning microscope (CLSM). A budding yeast cell was stained with three fluorescent stainings, red color of anti-GXM mAb specific for polysaccharide capsule followed by Alexa 568-conjugated goat anti-mouse Ig G (**a**), blue color of calcofluor white (CFW) specific for cellulose and chitin in fungal cell wall (**b**) and green color of Flu-P1 labeled r-javanicin (**c**). Merge of triple-fluorescent staining was represented (**d**). It was implied that r-javanicin localized in cytoplasm of *C. neoformans*. All photos were taken at the same magnification (scale bar = 2 µm).

**Figure 5 molecules-26-07011-f005:**
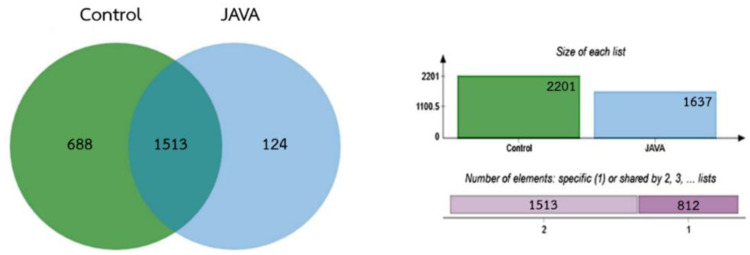
Venn diagram showing the number of *C. neoformans* proteins identified from untreated control group and r-javanicin treated group (0, 4, 8, 16, and 24 h), analyzed by LC-MS/MS. Left part shows the protein counts of shared (1513 proteins) and un-shared regions between yeast cell control (Control; 688 proteins) and javanicin treated *C. neoformans* (JAVA; 124 proteins). Charts on the top right show the size of protein clusters in control (green) and peptide treated group (blue) while the lower right is the cumulative numbers of shared (light purple) and un-shared proteins (purple) based on Venn diagram.

**Figure 6 molecules-26-07011-f006:**
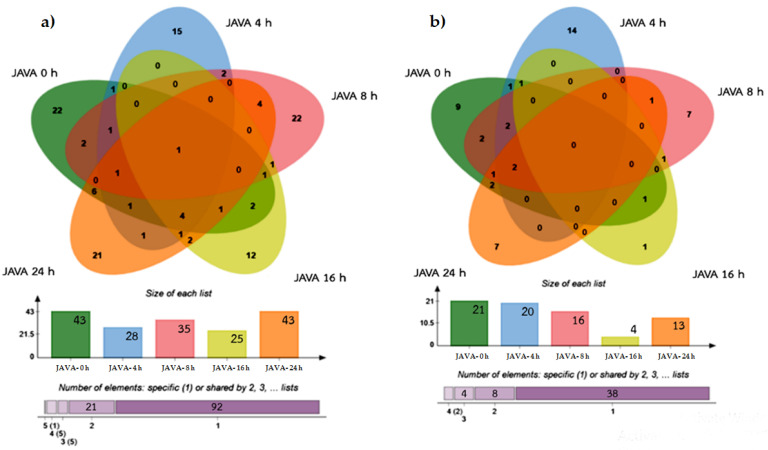
Venn diagram showing the proteins identified from *C. neoformans* in different time points after r-javanicin treatment in three biological replicates. The distribution of identified proteins in each condition (**a**) and the number of proteins classification based on protein function (**b**) are shown. Top part shows the protein counts of shared and un-shared regions of javanicin treated *C. neoformans* (JAVA) at 0, 4, 8, 16 and 24 h. Charts on the lower left and right show the size of protein clusters in each time point from 0 (green), 4 (blue), 8 (pink), 16 (yellow) and 24 h (orange), respectively. The bottom part is the cumulative numbers of specific (purple) and shared proteins (light purple) based on Venn diagram.

**Figure 7 molecules-26-07011-f007:**
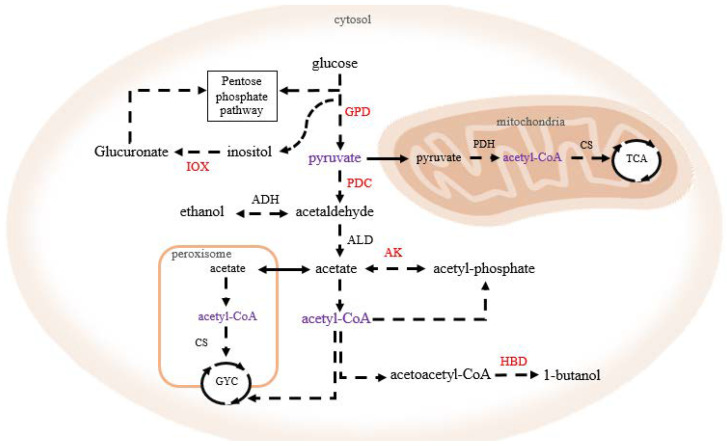
Proposed diagram of unique proteins involved in carbohydrate metabolism of *C. neoformans* in response to r-javanicin. Based on the proteomics analysis, many of fungal enzymes associated with carbohydrate catabolism after r-javanicin treated are identified (red letters) such as glyceraldehyde-3-phosphate dehydrogenase (GPD), pyruvate decarboxylase (PDC), 3-hydroxybutyryl-CoA dehydrogenase (HBD), inositol oxygenase (IOX), and acetated kinase (AK). Purple letters are the intermediate molecules. Some enzymes in the pathway including alcohol dehydrogenase (ADH), acetaldehyde dehydrogenase (ALD), citrate synthase (CS) and pyruvate dehydrogenase (PDH) are also indicated. TCA = Tricarboxylic acid cycle; GYC = Glyoxylate cycle.

**Table 1 molecules-26-07011-t001:** Antimicrobial effectiveness of r-javanicin against *C. neoformans* expressed in log CFU/mL reduction values.

MIC Values	Log Reduction (Average ± SD)
1 × MIC	4.78 ± 0.1601
2 × MIC	4.98 ± 0.0871
4 × MIC	4.93 ± 0.0455

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
