# Peer review of "Fungicidal Activity of Recombinant Javanicin against Cryptococcus neoformans Is Associated with Intracellular Target(s) Involved in Carbohydrate and Energy Metabolic Processes"

_molecules, 2021, doi:10.3390/molecules26227011_

Round 1

Reviewer 1 Report

This paper by Orrapin et al demonstrates an alternative antifungal agent. By employing microscopy and mass spectrometry techniques, the author shows that r-javanicin is able to inhibit against C-neoformans.

Reviewer questions:

  1. the authors should provide in the methods how analysis for the discovery proteomics was performed including the mass tolerance applied for the identification of proteins
  2. the authors should include the study limitations

Author Response

Dear, Reviewer 1       

Thank you very much for your kind suggestion and interesting of our manuscript. Your comments will be useful for improving of our research and manuscript preparation.  We are really appreciated. In a revised version of manuscript, words or phrases are changed according to reviewers’ recommendations. Some figures are either redraw or replaced according to reviewers’ suggestions. We would like to answer your questions as followed:

Reviewer questions:

1. The authors should provide in the methods how analysis for the discovery proteomics was performed including the mass tolerance applied for the identification of proteins

Response: We add the methods for discovery proteomics and the mass tolerance into the manuscript as reviewer’s suggestion (yellow colour).         

2. The authors should include the study limitations

Response: Thank you for your suggestion. As described in the manuscript, the major limitations of this study are:

  • Several expressed proteins in the late state (8-24 h) after peptide exposure are little known of function. Hence, we could not describe the overall anticryptococcal mechanism of r-javanicin.
  • Proteomics is a preliminary tool that guide us to identify the mechanism of action of r-javanicin against neoformans, therefore, supporting experiments need to be performed including the measurement of capsule thickness or determination of antifungal activity in capsular and acapsular strains in the presence of r-javanicin.

We have already included these limitations into the manuscript (yellow colour).  

Reviewer 2 Report

In this paper authors investigate the mode of action by which the recombinant antimicrobial peptide javanicin exerts its antifungal effect. Moreover, they explore the effect on several intracellular pathways by time course proteomic analysis. The paper underlines the importance of finding new therapeutic strategies to fight resistant fungal infections in humans.

Please find below some notes I would like the authors to take into account:

2-4 I am not sure the title authors chose perfectly aligns with the content of the paper. It is not only about the “ Proteomic Analysis of Cryptococcus neoformans in Response to  Recombinant Javanicin …”, but also exploring the mechanism of action of the peptide, by time-kill assay, SEM and fluorescent intracellular localization. Therefore, I suggest modifying the title to clarify the content to the readers.

55-59 Are there any human/mammalian defensins? I would recommend to implement the introduction with some references about human antimicrobial peptides directly involved in the immunity response, as authors are suggesting the use of antifungal molecules to tackle opportunistic fungal pathogens in patients.

126 How do authors measure or analyse morphological changes in SEM? Did they select any parameter, as the size of cells, diameter?

419 It is "effect" not "affect" of the peptide.

Supplementary figure S2: I find difficult the interpretation of the colour scale author chose for the different groups. I would recommend to choose colours with a stronger contrast  (for example too many shades of blue and the two shades of orange are difficult to discriminate).

I have just one concern. As authors say, a strong candidate which enter into clinical research should present antifungal properties and low cytotoxicity to human cells. Did they check toxicity on human cells (not tumoral cells) ? I think that this information is important in order to consider this defensin as a novel  therapeutic drug.

Author Response

Dear, Reviewer 2       

Thank you very much for your kind suggestion and interesting of our manuscript. Your comments will be useful for improving of our research and manuscript preparation.  We are really appreciated. In a revised version of manuscript, words or phrases are changed according to reviewers’ recommendations. Some figures are either redraw or replaced according to reviewers’ suggestions. We would like to answer your questions as followed:

Please find below some notes I would like the authors to take into account:

  1. 2-4 I am not sure the title authors chose perfectly aligns with the content of the paper. It is not only about the “ Proteomic Analysis of Cryptococcus neoformans in Response to Recombinant Javanicin …”, but also exploring the mechanism of action of the peptide, by time-kill assay, SEM and fluorescent intracellular localization. Therefore, I suggest modifying the title to clarify the content to the readers.

Response:        For clearly described, the title is rewritten according to reviewer’s recommendation (Fungicidal Activity of Recombinant Javanicin Against Cryptococcus neoformans is Associated with Intracellular Target (s) Involved in Carbohydrate and Energy Metabolic Processes).

  1. 55-59 Are there any human/mammalian defensins? I would recommend to implement the introduction with some references about human antimicrobial peptides directly involved in the immunity response, as authors are suggesting the use of antifungal molecules to tackle opportunistic fungal pathogens in patients.

Response: We apologize for possible unclear about defensins in line 55-59. They refer to plant defensins. Therefore, we have already included the human antimicrobial peptides involved in immunity response as reviewer’s recommendation (red letters).

  1. 126 How do authors measure or analyse morphological changes in SEM? Did they select any parameter, as the size of cells, diameter?

Response: In general, SEM is employed for visualization of outer surface morphological alteration of cells (bacteria, fungi, etc.) when the membrane targeted peptide is examined. In this study, the outer surface and also the size of yeast cells observed in recombinant javanicin treated and untreated control were compared whereas negligible difference with slightly shrinkage was observed in r-javanicin treated yeast cells as described in the manuscript.

  1. 419 It is "effect" not "affect" of the peptide.

Response: Thank you so much. We replace “affect” in line 419 with “effect” as reviewer’s suggestion (red letters).

  1. Supplementary figure S2: I find difficult the interpretation of the colour scale author chose for the different groups. I would recommend to choose colours with a stronger contrast (for example too many shades of blue and the two shades of orange are difficult to discriminate).

Response: We have changed the colours for better interpretation as reviewer’s suggestion.

  1. I have just one concern. As authors say, a strong candidate which enter into clinical research should present antifungal properties and low cytotoxicity to human cells. Did they check toxicity on human cells (not tumoral cells) ? I think that this information is important in order to consider this defensin as a novel therapeutic drug.

Response: Thank for your suggestion. The in vitro toxicity of recombinant javanicin has preliminary been determined using red blood cell hemolytic assay. The result indicated an increase in undesired hemolysis in a dose-dependent manner as our previously described (Orrapin et al., 2019). In order to reduce the cell toxicity, modification of some amino acid residues in the peptide chain, implementation of delivery system and combination of the peptide and conventional antifungal drugs for treatment (synergistic effect) should be performed.